# BACK TO FUNDAMENTALS: RE-EXAMINING MEMORIZATION IN DEEP LEARNING MODELS

## ABSTRACT

In supervised training, memorization is the ability of deep learning models to assign arbitrary ground truth labels to inputs in the dataset. Due to the computational difficulty of identifying existing memorized points, researchers often induce artificial memorization i.e, force the model to memorize the newly introduced points (via Noisy Label or Noisy Input). However, in this work, we show that this artificial *proxy* exhibits fundamentally different characteristics than the memorization real points (or natural memorization). To demonstrate this deviation, we re-examine two key findings derived from artificial memorization and compare them against natural memorization i.e., over-parametrization and increased training time increases memorization. We show that both these factors have the opposite effect i.e., they reduce natural memorization. Additionally, we find that memorization and train-test gap are *strongly* correlated (Pearson score 0.99). As a result, memorization is not necessary for generalization. Since real world models suffer from natural memorization (instead of the artificial one) our findings suggest the research community should focus on natural memorization, instead of the artificial proxy.

## 1 INTRODUCTION

Memorization is the ability of deep learning models to remember the point-label pairs in the training data (Zhang et al., 2017). The most precise method to check point memorization is through a leave-one-out test (Feldman & Zhang, 2020). Here, we train a model on the full data set and another model after having removed a single point from the data. If the model predicts a different label for the removed point then we mark it as memorized. However, if the model's prediction does not change, then we say that the model has generalized to the point (i.e., can classify the point correctly even if it is absent from the training data). This is repeated hundreds of times for each sample in the data set to account for the different sources of randomness (Feldman & Zhang, 2020), consequently yielding the memorization value of the point. One clear limitation of this approach is that it is prohibitively expensive. As a result, the procedure does not scale, even for small data sets such as CIFAR-100, which contains 50,000 training points. Therefore, studying *natural* memorized points in a real-world dataset incurs a significant computational cost.

To overcome this limitation, researchers employ proxy for natural memorization in the form of *artificial* memorization. Instead of identifying points that are memorized from the training data, researchers artificially introduce new points to the distribution, which are memorized by the model. This can be in two forms (Krueger et al., 2017; Collins et al., 2018; Stephenson et al., 2021; Morcos et al., 2018): (1) Noisy Label: An existing input from the training data is purposefully mislabeled (e.g., a cat mislabeled as APPLE) (2) Noisy Input: A completely unstructured point assigned an arbitrary label (e.g., picture of Gaussian noise mislabeled as APPLE). Since there is no relationship between these artificially introduced inputs and the corresponding label, the only way the model can predict the assigned label is via memorization. However, to get the models to memorize these points, they are over-trained using an unusually high number of iterations until the models predict the desired labels.

It is worth mentioning that there are several explicit examples of artificial memorization being used as a proxy. For instance, Zielinski et al. (2020b); Chatterjee (2020); Zielinski et al. (2020a); Cheng et al. (2021); Yao et al. (2019) use artificial memorization to build mechanisms to reduce model

memorization on artificial data. Only a small subset of papers use this approach. An interested reader can look into the space of reducing overfitting of noisy labelsSong et al. (2022), which is just a variation of artificial memorization.

However, this experimental framework rests on the assumption that artificial memorization is similar to natural memorization. Unfortunately, this assumption does not hold for four reasons: (1) Artificial memorization in the form of noisy input (i.e., Gaussian noise images) represents extreme outliers that might not occur in the data set. These noisy examples are completely unstructured, and will therefore have no overlapping features with the remaining points in the distribution. On the other hand, natural outliers in the distribution will at least have *some* overlapping features with remaining points. (2) Artificially memorized points characterized by noisy labels (i.e., mislabeled points) do not represent the majority of the dataset. Therefore, by focusing on them, we are overlooking far more important aspects of models memorizing natural, correctly labeled data. (3) Recent work has demonstrated that memorization does not only occur in the form of outliers and mislabeled points. Models can memorize points that belong to small-size sub-populations as well (Feldman & Zhang, 2020). For example, consider a data set consisting of 95 white cats and 5 black ones. A model trained on this data will likely memorize the five black cats in the dataset because there are so few representatives. (4) Finally, artificial memorization alters the normal training procedure. Specifically, it uses a significantly higher number of training iterations. Essentially, the model is trained until it correctly classifies (and, therefore memorizes) the artificial points. These additional iterations can be orders of magnitude greater than what is normally required for training a model (Arpit et al., 2017). Therefore, models trained on artificially memorized points diverge greatly from those in the real world. In light of these four reasons, it is clear artificial memorization is not a valid proxy for natural memorization. As a result, a natural question arises:

*Is artificial memorization a useful proxy for naturally memorized points?*

To answer this question, we evaluate whether findings from artificial memorization apply to natural memorization. We re-assess the two main claims proposed in the literature, i.e., given a fixed dataset, memorization increases as:

1. as the model size increases (i.e., over-parameterization) (Neel & Chang, 2023; Tirumala et al., 2022; Carlini et al., 2022; Thomas et al., 2020; Zhang et al., 2020; Bombari et al., 2022; Zhang et al., 2017; Tan et al., 2022; Ishida et al., 2020).

2. as training time (i.e., iterations) increases (Kandpal et al., 2022; Liu et al., 2020; Bai et al., 2021; Tanaka et al., 2018; Xia et al., 2020; Song et al., 2019).

We do so by identifying naturally memorized points from the training data using the recent algorithm from Feldman & Zhang (2020) and re-evaluating these findings. Our results demonstrate that the claims, that over-parameterization and increased training increase memorization, are flawed. We show that while increasing model parameters can increase the model's capacity to memorize data, it simultaneously improves the model's ability to learn the correct patterns from data, resulting in a potential decrease in memorization. As a result, as trainable parameters increase, natural memorization decreases.

Similarly, we show that the popular belief, that increasing training iterations always increases memorization, does not hold. We find that continued training, until the model reaches the minimum train-test gap, ultimately reduces memorization. Specifically, the rate of memorization is higher during the earlier epochs. As training continues, the number of memorized points starts to decrease as the model can learn features and generalize them. Given our refutation of the two major claims derived from artificial memorization, we call for a serious re-evaluation of existing literature in light of natural memorization to ascertain what does and does not hold true for real-world models.

Furthermore, during these experiments, we discover the phenomenon of *transient* memorization. This is when points are memorized under certain conditions but are generalized under others. We find that these can take two forms: **1) Model-Wise:** shallow models memorize points that are generalized by deeper models. Upon further investigation, we find that the transient memorized points consist of samples from smaller-sized sub-populations. Since shallow models do not have enough capacity to learn the rare patterns corresponding to smaller sub-populations, they instead memorize them in an attempt to classify them correctly. **2) Temporal-Wise:** points are memorized at ear-

lier epochs but are then generalized during the later epochs[1]. This happens because the model is not fully trained during the earlier epochs, and therefore, is unable to extract features from small sub-populations. However, with sufficient training, the model can learn these rarer patterns, thereby reducing memorization. As a consequence, improving the model's ability to learn rarer patterns will help minimize transient memorization.

In addition to reducing transient memorization, improving the model's ability to learn patterns also reduces the train-test gap. The research community has not reached a consensus on the relationship between memorization and the train-test gap, with some works claiming that the two have a direct relationship (Leino & Fredrikson, 2020; Salem et al., 2018; Yeom et al., 2018) and others showing the inverse (Hintersdorf et al., 2021; Kaya & Dumitras, 2021; Li et al., 2022; Feldman & Zhang, 2020). However, by studying natural memorization we show that as the train-test gap decreases, memorization decreases as well. This implies that techniques that improve test accuracy and reduce the train-test gap also decrease memorization. Our work makes the following contributions:

1. We re-evaluate two popular existing claims derived from the artificial memorization proxy i.e., over-parameterization and training iterations increase memorization. We show that these findings do not apply to natural memorization.

2. We identify the previously unknown phenomenon of transient memorization. This is when points are memorized under certain conditions but are generalized under others. We show that smaller sub-populations are mostly responsible for transient memorization. Therefore, improving the model's ability to learn rare patterns ultimately reduce memorization.

3. Lastly, we show that memorization and train-test gap are *strongly* correlated (Pearson score 0.99). As a result, we find that memorization is not necessary for generalization.

## 2 RELATED WORK AND MOTIVATION

Models can memorize entire data sets, not just a handful of points (Zhang et al., 2017). This is believed to be due to the over-parameterization of deep learning models (i.e., the model has more trainable parameters than training points) (Neel & Chang, 2023; Tirumala et al., 2022; Carlini et al., 2022; Thomas et al., 2020; Zhang et al., 2020; Bombari et al., 2022; Zhang et al., 2017; Tan et al., 2022; Ishida et al., 2020; Zhang et al., 2017). This was shown to be true for artificially memorized points where increasing the number of training parameters also increased the number of artificial points the model could memorize. The primary reason behind this behavior is that a greater number of parameters increases the model's expressivity, which enables the model to learn more intricate decision boundaries. As a result, the model can fit (and memorize) the artificially introduced points.

In addition to model complexity, another factor that is believed to increase memorization is the number of training iterations (Kandpal et al., 2022; Liu et al., 2020; Bai et al., 2021; Tanaka et al., 2018; Xia et al., 2020; Song et al., 2019; Zhang et al., 2020; Chatterjee, 2018; Zielinski et al., 2020b). While more useful features are learned during the earlier iterations (Arpit et al., 2017; Zielinski et al., 2020b), more memorization happens during the later ones. This is because as iterations progress, the model learns more complex decision boundaries, allowing it to fit the artificially introduced points. This resulted in the belief that models trained using high iterations would memorize more data, instead of learning useful features.

Having discovered the factors that increase memorization, the next natural question is if it is possible to reduce it while still being able to learn useful features. Since deeper models memorized more, the most obvious conclusion was the shallower models would memorize less. The intuition here is that shallower models are less expressive, learn simpler decision boundaries, and are therefore less prone to over-fitting the artificially introduced points. As a result, they will likely learn the underlying patterns in the data set, while being unable to memorize the artificial points. This realization led to a plethora of methods that attempted to combat memorization by merely simplifying the decision boundary, which included early-stopping (Liu et al., 2020; Bai et al., 2021; Tanaka et al., 2018; Xia et al., 2020; Song et al., 2019), dropout (Maini et al., 2023; Goel & Chen, 2021; Rusiecki, 2020; Xu et al., 2023), regularization (Cheng et al., 2021; Wei et al., 2020; Xu et al., 2022; Jiang et al., 2022; Yi et al., 2022), clustering (Stephenson et al., 2021), and at times entirely new frameworks (Han

---

[1]Transient memorization differs from over-fitting. This is because the former occurs during the earlier stages of training while the latter happens after the model has already been trained for a high number of epochs.

et al., 2018; Yao et al., 2020; Xie et al., 2021). However, all of this prior work uses artificial memorization. Therefore, the validity of these findings critically hinges on the same single assumption: that artificial memorization behaves the same way as natural memorization.

In this work, we show that this assumption is incorrect. We do so by re-evaluating the widely accepted beliefs about memorization by examining naturally memorized points. Specifically, we evaluate the belief that over-parameterization and increased training iterations cause memorization. We show that over-parameterization and increased training both reduce natural memorization.

## 3 IDENTIFYING NATURALLY MEMORIZED POINTS

However, before we re-evaluate these findings and show our results, we need to first describe how we identify the naturally memorized points. In this section, we describe the methodology we use, that was originally proposed by Feldman & Zhang (2020).

### 3.1 DEFINING MEMORIZATION

Central to our evaluation methodology is the definition of memorization. We adopt the one proposed by Feldman & Zhang (2020), arguably the most common, prominent definition in the memorization literature. Although other variations exist (Carlini et al., 2022; 2019; 2021), they are tailored to specific application domains (e.g., large language models), thus, we do not consider them in this work. According to Feldman & Zhang (2020), a point is memorized if it is correctly predicted only if it is present in the training data. Specifically, they provide a method to calculate the memorization score for each point in the training data. The score is the difference between the percentage of the models that classified the point correctly when it was *present* in the training data and the percentage of models that classified the point correctly when it was *absent* from the training data. It is worth mentioning that papers that employ artificial points are *implicitly* using Feldman & Zhang (2020)'s definition of memorization. This is because these artificial points have high scores as well.

Formally, consider a data point $x_i$ in the training set $S$ where $S = ((x_1, y_1)...(x_n, y_n))$. We train two sets of models $h$ on the data set $S$ using algorithm $A$. $(h \leftarrow A(S))$ are models where point $x_i$ are inside the training data. On the other hand, $(h \leftarrow A(S^{\setminus i}))$ are models where point $x_i$ is not inside the training data. The memorization score is the difference between the accuracy for point $x_i$ between the two sets of models:

$$\mathbf{Pr}_{h \leftarrow A(S)}[h(x_i) = y_i] - \mathbf{Pr}_{h \leftarrow A(S^{\setminus i})}[h(x_i) = y_i] \tag{1}$$

As we can see, the memorization score equation presented above captures the drop in accuracy once the point $x_i$ is removed from the training data. If the drop is larger than a threshold (set at 25% in the original paper), Feldman & Zhang (2020) marks that point as memorized. Overall, the definition captures the idea that a point $x_i$ is memorized if its prediction drops significantly if it is removed from the training data.

In simpler terms, we train 1000 instances of each of the models (i.e., 1000 models where $x_i$ is present and 1000 models where $x_i$ is absent from the training data ). We find that $x_i$ is classified correctly for 90% of the models when it is present in the training data (i.e., 900 out of the 1000 instances classified the point correctly). However, when $x_i$ is removed from the training data, its classification drops to 25% (i.e., 250 out of the 1000 instances classified the point correctly). The resulting memorization score is $90\% - 25\% = 65\%$. Since the memorization score (65%) is higher than the threshold (25% defined in the original paper), $x_i$ is marked as memorized. In contrast, if there is an insignificant change in the memorization score (i.e., $x_i$ is classified correctly, whether or not it is present in the training data) then we do not mark point $x_i$ as memorized.

Furthermore, if the memorization score is close to 100%, then it was only classified correctly when present in the training data. Therefore, this point belongs to a sub-population of size one (i.e., it is an outlier). If the score is closer to 0, then the point was classified correctly even if it was absent from the training data. This means that the point belongs to a large sub-population consisting of many points. In general, the lower the score, the larger the sub-population, and the larger the score, the smaller the sub-population (Abdullah et al., 2023). For example, in a dataset consisting of 100 white cats, 20 black cats, and 1 purple one, the white cats will have memorization scores closer to 0, the purple cat will have a one closer to 1, and the black cats will have one somewhere in between.

| CIFAR10/100 | | | | | | |
|---|---|---|---|---|---|---|
| | SmallVGG | MedVGG | LargeVGG | VGG19 | Resnet18 | Resnet50 |
| Parameters | 0.5M | 1M | 7.5M | 20M | 11M | 24M |
| Train Accuracy | 98.14 / 97.87 | 99.79 / 99.89 | 100.0 / 99.98 | 99.98 / 99.96 | 100.0 / 99.98 | 100.0 / 99.98 |
| Test Accuracy | 87.25 / 59.35 | 88.85 / 62.89 | 90.46 / 67.80 | 92.13 / 68.23 | 93.58 / 73.41 | 93.78 / 75.17 |

Table 1: To evaluate the relationship between natural memorization and over-parameterization, we trained the following models with varying number of trainable parameters. The accuracy is in the format **CIFAR10 Accuracy / CIFAR100 Accuracy**.

| Tiny ImageNet | | | | |
|---|---|---|---|---|
| | Resnet18 | Resnet50 | Vit-Tiny | Vit-Small |
| Parameters | 11M | 24M | 5M | 22M |
| Train Accuracy | 100.0 | 100.0 | 100.0 | 100.0 |
| Test Accuracy | 57.29 | 62.02 | 70.20 | 79.3 |

Table 2: To evaluate the relationship between natural memorization and over-parameterization, we trained the following models with varying number of trainable parameters. The accuracy is in the format **Tiny ImageNet Accuracy**.

## 3.2 IDENTIFYING MEMORIZED POINTS

So far, we have only defined how to calculate the memorization score in Equation 1. However, calculating the actual score is a much harder task due to its computation complexity. This is because it requires running the classic leave-one-out technique, which is comprised of the following steps: 1) train a model on the entire training data 2) remove a single point from the data 2) retrain the model on the remaining data 3) check if the removed point is correctly classified. 4) repeat steps 1-3 a few hundred times to account for the different sources of randomness introduced during training (e.g., the varying initialization, GPU randomness, etc.). 5) calculate the memorization score across the hundreds of runs. 6) repeat all the above steps for each point in the training data. It is painfully obvious that using the leave-one-out methodology to calculate the memorization score requires the user to train hundreds of thousands of models. Therefore, running this experiment over even a small dataset (such as CIFAR-100 which contains 50,000 training points) will require a large amount of resources and is, therefore, computationally intractable.

To overcome this limitation, Feldman & Zhang (2020) develop a technique to *approximate* the memorization scores. Instead of removing one point at a time, the authors randomly sample a fraction $r$ of the points from the training set (originally of size $n$) and leave the remaining points out of training. The number of points used in training is then $m = r \cdot n$, $0 \leq r \leq 1$. In Feldman & Zhang (2020) the authors use $r = 0.7$ for their experiments. The authors repeat this $k$ times. The exact value of $k$ depends on the dataset but is typically on the order of a few thousand models. As a result, a random point $x_i$ will be present in approximately $k \cdot r$ of the total trained models and will be absent from $k \cdot (1 - r)$ of them. By aggregating the results over both sets of models, the authors can approximate the memorization score for $x_i$. All the points that have a higher memorization score than some predetermined threshold (specified in the original work as 25%) are said to be memorized.

Running this methodology will help us calculate the memorization scores of the points in the dataset. With these scores, we can identify natural memorization: the pre-existing points that are being memorized by the model.

## 4 EXPERIMENTAL SETUP AND RESULTS

Having described how we extract the naturally memorized points, we now re-examine the role of over-parameterization and training iterations on natural memorization. While doing so, we discover the phenomenon of transient memorization. Finally, we will explore the relationship between memorization and the train-test gap.

To unearth the relationship between natural memorization and over-parameterization, we train a series of models with an increasing number of trainable weights. We designed four models using

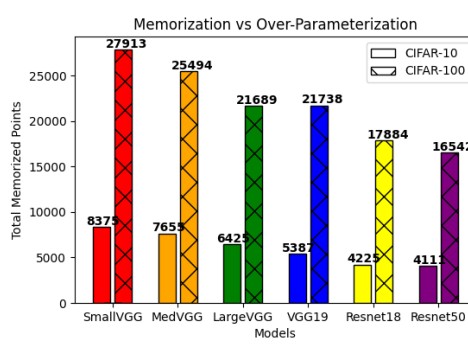 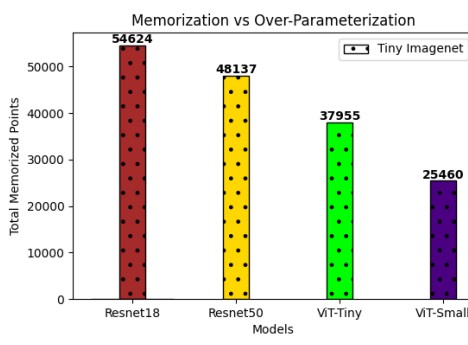

(a) CIFAR-10/100          (b) Tiny ImageNet

Figure 1: The figure shows the relationship between memorization and model complexity, across different model families. When comparing VGG models, we can see that as number the of parameters increases, memorization decreases. To validate our findings, we reproduce the experiment across the RESNET and the transformer based ViT family of models. This is stark contrast to artificial memorization, which increases as model complexity increases.

VGG blocks (SmallVGG[2], MedVGG, LargeVGG, VGG19), two model types using the Resnet architecture (Resnet18 and Resnet50), and three Vision Transformers (ViT Tiny and Small) shown in Tables1 and 2. Here, the goal is to observe how varying the trainable parameters, while keeping the architecture family constant, impacts memorization. We repeat the experiment on two different architectures (VGG and RESNET) to see if the findings hold. Observing the total number of naturally memorized points for each architecture will help reveal the role of over-parameterization on memorization. Next, to understand the role of training iterations, we calculate the number of memorized points at each iteration of training. This reveals how memorization varies across training.

We identify the naturally memorized points using the method in Section 3. We use a similar training setup to the one outlined in Feldman & Zhang (2020). Specifically, we train the models for 100 epochs, using a batch size of 512, with a triangular learning rate of 0.4. However, based on recent work (Abdullah et al., 2023), we make one minor modification and use weight decay to avoid undertraining. We train 2,000 models for each architecture mentioned in Table 1, use data augmentation of Random Horizontal Flip and Random Translate, and use Equation 1 to identify all the naturally memorized points. We repeat these experiments pre-trained ViTs [3]. We conduct this experiment using three training data sets (CIFAR-10, CIFAR-100, and Tiny ImageNet) to get a better understanding across different data sets.

This experimental framework gives us two advantages. First, it allows us to study natural memorization, which is experienced by real-world models. Second, unlike artificial memorization which requires very high training iterations (explained in the Section 1), we simply train the models to their maximum test accuracies. As a result, contrary to models trained on artificial points, our training setup closely resembles what is used in the real world.

### 4.1 Over-parameterization vs Natural Memorization:

Figure 1[4] shows that as over-parameterization increases, memorization decreases. For example, SmallVGG memorized more points than VGG19, even though VGG19 has significantly more trainable weights. We observe a similar trend when comparing the Resnet models and ViT models where deeper models memorize less than shallower ones. This shows that shallow models learn fewer useful features and are forced to memorize points to classify them correctly. On the other hand, the deeper models can learn more useful features and are therefore able to classify points correctly

---

[2]Details of the architecture are provided in the Appendix

[3]https://huggingface.co/timm

[4]We repeat these experiments without data-augmentation and weight decay and show that are results do not change (Appendix Figure 6)

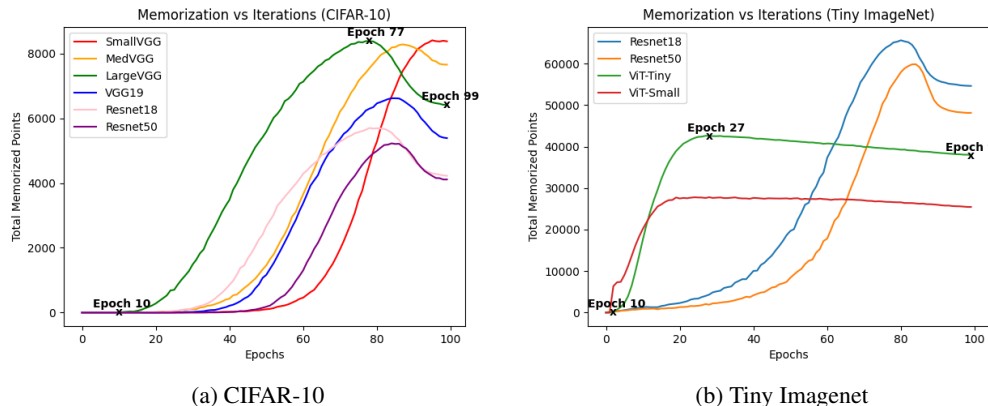

(a) CIFAR-10                                             (b) Tiny Imagenet

Figure 2: The figure above shows the relationship between memorization and training iterations. CIFAR-100 results in the Appendix (Figure 5).

without memorization. This means that deeper models have a higher learning capacity than shallow models. As a result, increasing over-parameterization decreases memorization.

In addition to a higher learning capacity, our results show that deeper models have a higher memorization capacity as well. Specifically, shallow models (like SmallVGG) have a lower training accuracy than deeper models (like VGG19) (Table 1). This means that SmallVGG is unable to fit certain points that can be fit by VGG19. Upon further inspection, we saw that points misclassified by SmallVGG were then either memorized or generalized by the larger VGG19 model. Since smaller models are unable to fit the entire training data, this means that smaller models lesser capacity to memorize *and* lesser capacity to generalize in comparison to the larger models.

Our results show that the belief that over-parameterization increases model memorization does not apply to natural memorization. We can see that over-parameterization and memorization have a nuanced relationship. Increasing parameters increase memorization capacity. However, over-parameterization also improves the model's learning capacity, thereby reducing the total number of memorized points.

## 4.2 Training Iterations vs Natural Memorization:

We can observe a similarly nuanced relationship between memorization and training epochs in Figure 2. Memorization can be split into three different stages.

**Stage 1:** This initial stage consists of the first few epochs and is characterized by the unique *absence of any memorization*. For example, consider the LargeVGG plot (green line) in Figure 2a. There is no memorization from Epoch 0 to Epoch 10. This can be observed across all the models and data sets. This reaffirms the observations by prior work, which shows that the model learns *easy* samples during the first few epochs (Arpit et al., 2017). **Stage 2:** During this stage, we start to observe a gradual *increase in memorization*. We can see that for LargeVGG (green line) memorized points increase from zero at Epoch 10 to 8,300 at Epoch 77. **Stage 3:** The final stage is characterized by a *reduction in memorization*. Points that were memorized at the earlier epochs are generalized to during the final epochs. For example, the number of memorized points falls from Epoch 77 onwards (Figure 2a). Memorization reduces from approximately 8,300 to 6,400 points from Epoch 77 to 99. This observation holds true even for the ViT models, even if it is more subtle. We believe that this is due to the use-pretrained base models that help achieve high test accuracy in the first few epochs, and the marginal improvements in accuracy for the remaining training cycle.

Our results show that the belief that increased training iterations increase model memorization does not apply to natural points. We show that the number of iterations has a more nuanced impact on memorization. While there is little to no memorization during Stage 1, memorization starts increasing and reaches its peak during Stage 2, followed by a stark reduction during Stage 3. This means, that if the model is trained for long enough, the memorization rate will eventually fall. These results are somewhat similar to *epoch wise* double descent (i.e., longer training regimes can correct

over-fitting (Nakkiran et al., 2021)) and thus a validation for our work. However, there is one main difference between the two. Double descent experiments have demonstrated that it takes thousands of additional epochs for the model to correct the over-fitting of these points, far beyond a normal training regimen of real-world models. On the other hand, we show that memorization of natural points can be corrected within the first hundred epochs, and thus applicable to real-world models.

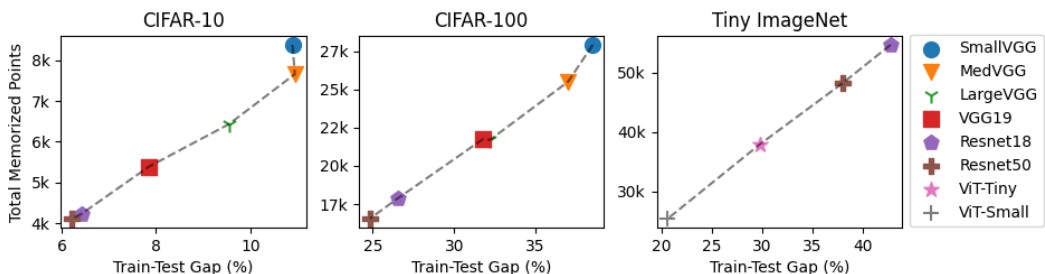

Figure 3: The relationship between memorization and the train-test gap. We can see across all three datasets, memorization and train-test gap are *strongly* correlated (Pearson Score 0.99).

### 4.3 EXPLORING TRANSIENT MEMORIZATION:

One interesting behavior that has been exposed by our experiments is that of *transient* memorization i.e., points that are memorized under certain conditions, but are then subsequently learned. These take two forms: **Model-Wise**: points are memorized by shallow models but are generalized by deeper ones and **Temporal-Wise:** points are memorized in the earlier epochs but then generalized to the later ones. We call this phenomenon *transient* memorization. These are points that are memorized under certain conditions but are generalized under others. In this subsection, we explore this phenomenon and identify what type of points are prone to behavior.

**Model-Wise:** To do so, we first identify the model-wise transient points that were memorized by the smaller models but learned by the larger ones. Specifically, SmallVGG and Resnet50 memorized 8,375 and 4,111 points from CIFAR-10 datasets, respectively. This means there are approximately $8375 - 4111 = 4264$ transient memorized points (i.e., memorized by the SmallVGG but were then learned by Resnet50). Having identified the transient points, we observe their memorization scores calculated using Equation 1. As a reminder to the reader, points from larger sub-populations have lower scores, and ones from smaller sub-populations have a higher score (Section 3.1). Our calculations show that while the average memorization score for the entire data set is 11.17% ± 19.13%, the transient points have an average score of 44.99% ± 15.02%. This shows that the model-wise transient points do not consist of outliers (as the score is not close to 100) nor do they consist of sub-populations with many points (as the score is not close to 0). This means that transient points consist of samples from smaller-sized sub-populations, with rarer occurrences in the training data. As a result, these are difficult for shallow models to learn. Therefore, the only way shallow models can classify these smaller sub-populations correctly is via memorization. However, as the model depth increases, the capacity to learn rare patterns also improves. As a result, the model can learn better features that help it classify the smaller sub-populations correctly without memorization.

Since results can be impacted by arbitrary cut off values for the memorization scores, we also present the results of the distributions of memorization scores across different models in Figure 4(a). We can see that as we increase model complexity and architecture, memorization decreases. Thereby showing, that increasing complexity reduces natural memorization memorization.

**Temporal-Wise:** We see a similar trend in temporal-wise transient memorization across epochs i.e., points that are memorized in the earlier epochs but are learned in the later ones. To do so, we compare the points memorized at the Epoch with the highest memorization (Epoch 77) against the one with the lowest memorization (Epoch 99). There are 8380 and 6425 points memorized in Epochs 77 and 99 in Figure 2. There are approximately 1955 transient memorized points. These points have an average memorization score of 35.71% ± 4.69%, which is larger than the average score across all the points in the training data of 11.17% ± 19.13%. This demonstrates the transient memorized points consist of smaller sub-populations, which is similar to our earlier observation. Therefore,

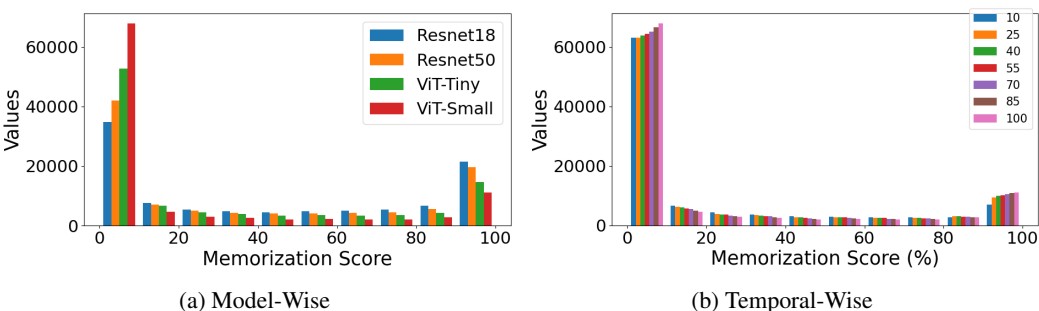

(a) Model-Wise

(b) Temporal-Wise

Figure 4: Two forms of transient memorization on the Tiny ImageNet Dataset. **(a) Model Wise:** Memorization across model architectures. **(b) Temporal-Wise:** Memorization across epochs.

as the number of epochs increases, the model's ability to learn rare patterns improves. Therefore, the features associated with the smaller sub-populations are learned by the model, thereby reducing reliance on memorization. Additionally, we note that the memorization score of the transient points across epochs is lower than the score across models ($35.71\% \pm 4.69\%$ vs $44.99\% \pm 15.02\%$), this means that the transient points across epochs belong to slightly larger sub-populations. However, since the score is still larger than the average, it means that the transient points on average belong to smaller sub-populations.

Finally, we show the results of the distributions of memorization scores across different epochs for the Tiny ImageNet dataset trained on ViT-Small in Figure 4(b). Temporal-wise memorization is *visually* subtle (compared to model-wise), but present nonetheless. Specifically, increasing training iterations generally reduces memorization scores. The only exception is that higher iterations lead to points with extremely high memorization scores (between 90%-100%). At the same time, we can observe for most memorization scores (in the range 10%-90%), lower iterations produce high scores. This reinforces our argument that increasing iterations can, in general, reduce natural memorization.

### 4.4 NATURAL MEMORIZATION VS TRAIN-TEST GAP:

Our results show that transient points consist of smaller-sized sub-populations. To enable the model to learn features associated with these samples, instead of memorizing them, we can increase model complexity and/or training iterations. Interestingly, this also helps the model's test accuracy, thereby reducing the train-test gap. However, as we showed earlier, the research community is torn on the matter of memorization and the train-test gap. While some works claim a direct relationship i.e., the train-test gap increases memorization (Leino & Fredrikson, 2020; Salem et al., 2018; Yeom et al., 2018), others show the inverse (Hintersdorf et al., 2021; Kaya & Dumitras, 2021; Li et al., 2022; Feldman & Zhang, 2020). Now, we re-evaluate this notion by studying natural memorization.

Figure 3 shows that there is a strong relationship (Pearson score 0.99) between the train-test gap and memorization. This is because smaller models are unable to learn features corresponding to the smaller sub-populations, resulting in lower test accuracy, a larger train-test gap, and therefore, greater memorization. In contrast, large models can learn the rarer patterns in the data, resulting in a higher test accuracy, smaller gap, and therefore, lesser memorization. This explains why VGG19 memorized a significantly higher number of points than Resnet18, even though VGG19 has twice as many parameters (Table 1). This is because Resnet18 includes architectural improvements to increase the model's ability to learn rarer patterns thereby increasing test-set accuracy and reducing the model's reliance on memorization.

## 5 DISCUSSION AND TAKEAWAYS

**Artificial and Natural Memorization through Feldman & Zhang (2020)** As mentioned previously, many existing works studying artificial memorization *implicitly* use Feldman & Zhang (2020)'s definition of memorization. This means that the core issue does not lie in the notion of memorization or its definition (since both artificial and natural memorized points have high scores).

Instead, it lies in the fact that prior works incorrectly assume that their findings from the artificial proxy would translate to natural memorization. As a result, future researchers are encouraged to study memorization from the natural lens, instead of using the artificial proxy. To add to that, this definition of memorization only applies to classification problems. One promising direction for future work is to see if these findings can be extended to LLMs.

**Train-test gap and Memorization are *strongly* correlated:** We can observe in Figure 3 that the increasing model size reduces memorization while simultaneously increasing test accuracy, with a Pearson correlation score of 0.99. This strong correlation alludes to the fact that generalization and memorization are inversely proportional: As test accuracy increases, memorization decreases. Intuitively, when a point is memorized, removing it from the training data results in an incorrect classification. However, deeper models have the ability to learn better features. As a consequence, they are able to learn the features needed to classify the point even when it is absent from the data set. One way to track how quality of features learned by the model is its test accuracy (higher accuracy, better features). Therefore, as the model learns better features, it memorizes fewer points, while simultaneously generalizing better to test points. This is further explained by the results of the shallow models. Specifically, since shallow models lack the ability to learn robust features. As a consequence, when a point is removed from the training data, it will likely be misclassified. Therefore, shallow models will likely memorize points. Therefore, one simple way to minimize memorization is to train the model to the lowest train-test gap.

**Memorization is not necessary for generalization** Feldman & Zhang (2020) argue that memorization is *necessary* for generalization. This means that a model's ability to perform well on the test set is predicated on its ability to memorize the small-subpopulations on the distribution's long tail. However, we find that memorization is *not* necessary for generalization. This is because 1) small-subpopulations experience transient memorization and are learned by increasing model complexity and training epochs. 2) As transient memorization decreases, generalization increases, and train-test gap decreases. Thus, better feature learning, not memorization, drives performance. Therefore, reliance on memorization is a fallback mechanism for less capable models (explaining their poor test-set performance), while models with greater learning capacity and training regimes demonstrate that generalization, not memorization, is the key to achieving high accuracy.

## 6 CONCLUSION

The study of memorization has been based on the premise that artificial memorization is a valid proxy for natural memorization. In this work, we show that this is not the case. We do so by evaluating two of the most popular beliefs from artificial memorization i.e., model complexity causes memorization and high training iterations cause memorization. We show that these do not apply to natural memorization. Additionally, we discover the previously unknown phenomenon of transient memorization. This is when points are memorized under certain conditions but are generalized to under other conditions. Given our experimental results, we challenge the idea that artificial memorization might always be a good proxy for natural points. Future work should focus on further experimentation to validate this finding across more domains and architectures. In light of our findings, researchers are encouraged to use natural memorization instead of using the artificial proxy.

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

## A    MODEL ARCHITECTURES:

Below we provide information about the custom VGG models built using the standard VGG block. We use the *VGGNumber* $\rightarrow$ *MaxPool* to describe the architecutre where the *VGGNumber* represents the size of the standard VGG Block.

**VGGsmall:** $64 \rightarrow MaxPool \rightarrow 64 \rightarrow MaxPool \rightarrow 64 \rightarrow MaxPool \rightarrow 64 \rightarrow MaxPool \rightarrow 512 \rightarrow MaxPool \rightarrow FC$.

**VGGmed:** $64 \rightarrow MaxPool \rightarrow 128 \rightarrow MaxPool \rightarrow 128 \rightarrow MaxPool \rightarrow 128 \rightarrow MaxPool \rightarrow 512 \rightarrow MaxPool \rightarrow FC$.

**VGGlarge** $64 \rightarrow MaxPool \rightarrow 512 \rightarrow MaxPool \rightarrow 512 \rightarrow MaxPool \rightarrow 512 \rightarrow MaxPool \rightarrow 512 \rightarrow MaxPool \rightarrow FC$.

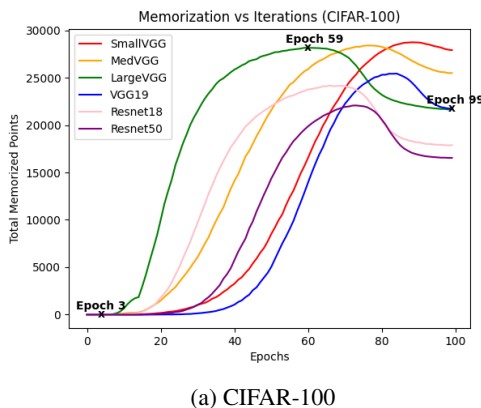

(a) CIFAR-100

Figure 5: The figure above shows the relationship between memorization and training iterations.

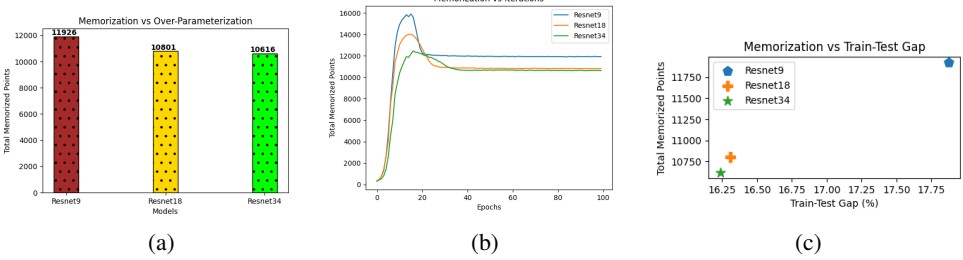

(a)  (b)  (c)

Figure 6: To disentangle the impact of the training regime from memorization, we repeat our experiments without data augmentation and weight decay. As we can see, our findings do not change from the earlier results. a) Increasing model complexity decreases memorization b) Increases iterations decreases memorization c) Memorization and Train-Test gap are strongly correlated.