# OpenReview forum: "Back to Fundamentals: Re-Examining Memorization in Deep Learning Models"
_ICLR.cc/2025/Conference — Submitted to ICLR 2025_

### Official Review · Reviewer_SHMd · 2024-10-30

**Soundness:** 2
**Presentation:** 2
**Contribution:** 2
**Rating:** 3
**Confidence:** 3

**Summary:**

The paper discusses the definition and the measurement of memorization in the previous work. It states that a number of papers in the literature measure memorization for artificial data points introduced in the dataset. The paper discusses that the conclusions made for the artificial data points do not apply to the natural data points in the training dataset. Specifically, the paper states that the claims made in the literature about memorization on model size and training time do not hold for the natural training data points. The authors ground those claims based on the experimental results for VICReg and ResNet models trained on CIFAR10 and CIFAR100 datasets.

**Strengths:**

+ The paper questions important aspects of memorization definition and measurement. It discusses the memorization measurement for natural data points vs data points that are artificially introduced via noisy labels or noisy inputs.
+ The paper cites and discusses a number of important works on memorization in literature.
+ Transient memorization proposed by the authors  is an interesting phenomenon.

**Weaknesses:**

+ The story of the paper is not very clear. More specifically, the paper relies on Feldman and Zhang’s measurement of memorization and states that transient memorization happens for smaller sub-populations. The smaller sub-population, however, could be the long tail. This contradicts the study proposed in Feldman and Zhang. Feldman and Zhang specifically discuss that long tail is being memorized for the final checkpoint not for intermediate epochs.
+ Overall the experiments for the artificial data points are missing. The paper doesn't give the full picture and doesn't show clear differences between natural and artificial data points based on the experimental results. Currently the experiments are conducted only for the natural data points.
I'd suggest running experiments for artificially inserted / noisy labels and inputs in CIFAR and show that for VGG and ResNet memorization is indeed different for those data points as opposed to the natural data points. Previous literature might have a different experimental setup. The authors should run the experiments for their experimental setup and datasets when comparing with their approach.

+ The study of relationship between model size and number of epochs doesn't seem very novel. There is work on that especially on the large language models.
E.g. Carlini, et.al. discuss that the memorization increases with the capacity of the model
https://arxiv.org/pdf/2202.07646

**Questions:**

1. There are papers that study the relationship between model's size and the training dynamics with respect to memorization (especially large language models ). Have you considered looking into that ? Are the claims made here true only for vision classifiers ?

---

> ### Author Response · Authors · 2024-11-19
>
> > This contradicts the study proposed in Feldman and Zhang.
>
> We thank the reviewer for this observation. Yes, we agree: Our findings do contradict Feldman and Zhang. Our experiments are much more extensive that theirs. We evaluated memorization across different architectures and across epochs. In contrast, Feldman and Zhang's seems limited compared to ours. For example, they do not evaluate over ViT based models, that have much higher model accuracy. This provides motivation to re-evaluate their findings using the same meticulous experimental setup we used in our work.
>
> > The study of relationship between model size and number of epochs doesn't seem very novel
>
> Our findings *contradict* those found in the reviewer cited paper. The paper claims memorization increases with capacity, while we show it decreases with capacity. These contradictory findings across domains is even more reason for the community to be aware of our work.
>
> > Are the claims made here true only for vision classifiers ?
>
> The Feldman and Zhang definition of memorization is limited to classification problems. LLM pre-training is usually unsupervised. Therefore, this definition will not apply. While it would be interesting to see if our observations still hold for LLMs, it is outside the scope of the current work.

---

> ### Comment · Reviewer_SHMd · 2024-11-24
> **Response to authors**
>
> Thank you for reading the feedback and answering my questions.
> I understand that you experimented with more models that Feldman and Zhang but if your findings contradict Feldman and Zhang then it should be made clear in the story of the paper. The paper comes across that you reject the claims made for artificially inserted examples for natural examples. Meaning, the claims made for artificially inserted examples do not apply to natural examples. The paper does not discuss the weaknesses of the Feldman and Zhang and your contributions beyond that work.

---

> ### Author Response · Authors · 2024-11-24
>
> Thank you for the feedback. We have updated the story of the paper to reflect your feedback. We modified the abstract, intro, and discussion section to show how our findings contradict Feldman et al.
>
> Please let us know if you have any other suggestions on how we can improve out work.

---

> ### Comment · Reviewer_SHMd · 2024-11-25
> **Response to authors**
>
> Thank you for the quick response.
> Overall from the current standpoint my concerns about the experimental setup for the artificially added datapoints remain open. I think that it might be better to have a more thorough experimental setup for artificially added vs natural datapoints.
> Also the paper doesn't add any novelty in terms of methods and algorithms. The main novelty comes from the empirical findings based on existing methods. At the same time the findings are a bit hard to generalize. It might be better to strengthen empirical evidence of the paper and discuss why these finding makes sense in a general use case in theory.
>
> I'll leave my score unchanged.

---

> > ### Author Response · Authors · 2024-11-25
> >
> > > At the same time the findings are a bit hard to generalize. It might be better to strengthen empirical evidence of the paper and discuss why these finding makes sense in a general use case in theory.
> >
> > Thank you for your valuable feedback. If there is any specific experiment that would help support (or reject) our findings, please feel free to share. We would be happy to run them in order to strengthen our work.

---

> > > ### Author Response · Authors · 2024-11-30
> > >
> > > Dear Reviewer,
> > >
> > > As we near the discussion deadline, please let feel free to let us know if you have any more questions or suggestions.

---

### Official Review · Reviewer_C2wk · 2024-10-30

**Soundness:** 2
**Presentation:** 2
**Contribution:** 2
**Rating:** 6
**Confidence:** 4

**Summary:**

This paper re-evaluates memorization in deep learning by distinguishing "natural memorization" (memorization of real data) from "artificial memorization" (induced with noisy data or labels), arguing that the latter fails as an accurate proxy for real-world behaviors. The authors find that over-parameterization and longer training reduce natural memorization, as larger models learn complex patterns and generalize better over time. They also introduce "transient memorization," where certain data points shift between memorization and generalization depending on conditions like model size or training stage. By exploring the relationship between train-test gaps and memorization, the study suggests that models with smaller train-test gaps tend to memorize less.

**Strengths:**

**1. Novelty:** The paper challenges the existing reliance on artificial memorization by demonstrating significant differences between artificial and natural memorization. This is valuable, given the application of artificial memorization as a proxy in prior studies.

**2. Results on Over-Parameterization:** The finding that increased model complexity can reduce memorization, contrary to popular belief, is useful. It aligns with ongoing discussions about the balance between capacity and generalization, especially given the risks of overfitting in high-parameter models.

**Weaknesses:**

**1. Transient Memorization as a New Phenomenon:** I believe the authors have grouped two distinct phenomena into one category, which reduces its impact

 a. Dynamics-wise: This refers to the authors' claim that they observe a new phenomenon where certain data points are memorized and then generalized. While forgetting during training is already known (see Toneva et al., 2018), the evidence presented here is novel enough to stand as a new insight.

 b. Model-wise: The authors suggest that "since shallow models do not have enough capacity to learn the rare patterns corresponding to smaller sub-populations, they instead memorize them in an attempt to classify them correctly." However, the claim that larger models memorize less due to their improved learning capacity is too generic to be presented as novel.

To strengthen the paper, I recommend splitting transient memorization into these two distinct forms. The model-wise transient memorization could be more appropriately discussed when exploring the role of over-parameterization.

**2. Insufficiency of Memorization Thresholds:** The fixed memorization threshold of 25% is applied across various experiments without sufficient justification. This raises concerns about arbitrary cutoffs potentially affecting results, especially when claiming that natural memorization behaves differently across sub-populations. While I believe the results are fundamentally correct, presenting a histogram showing the distribution of memorization scores would add significant value. For instance, memorization scores are often bimodal [2], with many samples showing either zero or high scores. A histogram representation would illustrate how this bimodality changes (a) as training progresses and (b) as model size changes, thereby substantially strengthening the paper.

**3. Train-Test Gap and Memorization:** The paper proposes that reducing the train-test gap decreases memorization. However, the empirical support for this claim is limited to two architecture families (VGG and ResNet) tested on CIFAR datasets. While I understand that using ImageNet would be computationally prohibitive, exploring other tasks (e.g., language or segmentation tasks) could provide stronger evidence for the generalizability of this finding.

[1] Toneva, Mariya, Alessandro Sordoni, Remi Tachet des Combes, Adam Trischler, Yoshua Bengio, and Geoffrey J. Gordon. "An empirical study of example forgetting during deep neural network learning." arXiv preprint arXiv:1812.05159 (2018).

[2] Michal Lukasik, Aditya Krishna Menon, Ankit Singh Rawat, Vaishnavh Nagarajan, and Sanjiv Kumar. "What Do Large Networks Memorize?" 2023. Accessed October 30, 2024. https://openreview.net/forum?id=QcA9iGaLpH4.

**Questions:**

1. Could transient memorization be split into distinct dynamics-wise and model-wise categories to better capture its nuances? Specifically, does collapsing both dynamics-wise and model-wise memorization under the umbrella of transient memorization risk obscuring important insights about model behavior?

2. Could architectural variations other than ResNet and VGG affect the train-test gap and memorization dynamics? For instance, do transformer-based architectures show similar relationships, or might they reveal new insights regarding the impact of model complexity on memorization? Mostly because transformers are hard to train on small datasets, thus pre training may have a significant effect on the results.

---

> ### Author Response · Authors · 2024-11-19
>
> We thank the reviewer for their comments. To address them, we made changes to the paper to reflect the two distinct types of memorizations, added figures with memorization score distributions, and carried out additional experiments over Tiny ImageNet dataset using ViT and Resent based models. This has improved the quality of work. If the reviewer has any recommendations or questions, please feel free to let us know.
>
> > 1. Transient Memorization as a New Phenomenon:
>
> We thank the reviewer for this point. We went back and updated the paper to describe the presence of two distinct forms of memorization.
>
> > However, the claim that larger models memorize less due to their improved learning capacity is too generic to be presented as novel.
>
> We agree that this might not be novel on its own. However, considering that others papers have made a contrary claim (via use of artificial memorization), we believe that it is important for the community to be aware of how memorization really behaves in ML models.
>
> > 2. Insufficiency of Memorization Thresholds:
>
> This is a good point. We updated our paper to show the distribution of memorization scores across memorization scores (Figure 4) and updated the related text.
>
> Figure 4a) We found that model-wise transient memorization is easily discernable, with shallower models memorizing less.
>
> Figure 4b) Similarly, temporal-wise transient memorization is more subtle, but present nonetheless. Specifically, increasing training iterations generally reduces memorization scores. The only exception is that higher iterations lead to points with extremely high memorization scores (between 90%-100%). At the same time, we can observe for most memorization scores (in the range 10%-90%), lower iterations produce high scores. This reinforces our argument that increasing iterations can, in general, reduce natural memorization.
>
> > 3. Train-Test Gap and Memorization
>
> To address this concern, we repeated our experiments on a more complex dataset, Tiny ImageNet, which has 200 classes over 100,000 images using both ViT based and Resnet based architectures. We observed that the our original finding, that Train-Test Gap and Memorization are strongly correlated, still holds (Pearson score 0.99). This showed that our results are independent of model architecture and pertaining. The reviewer can find the results in Figure 3.

---

> > ### Comment · Reviewer_C2wk · 2024-11-22
> >
> > Thank you for your response, I have read the updated paper and the rebuttal comments.

---

> > > ### Author Response · Authors · 2024-11-25
> > >
> > > Dear Reviewer C2wk,
> > >
> > > Thank you for your valuable feedback. We have provided detailed responses to your questions and hope they address your concerns thoroughly. As the interactive discussion period is coming to a close, please don’t hesitate to reach out if you need further clarification or additional information.

---

> > > > ### Author Response · Authors · 2024-11-30
> > > >
> > > > Dear Reviewer,
> > > >
> > > > As we near the discussion deadline, please let feel free to let us know if you have any more questions or suggestions.

---

### Official Review · Reviewer_JPiP · 2024-10-31

**Soundness:** 2
**Presentation:** 3
**Contribution:** 2
**Rating:** 3
**Confidence:** 3

**Summary:**

The paper considers memorization in DNNs. Going back to the original definition via expensive leave-one-out trials, the authors question whether artificial memorization experiments as proxies for 'natural memorization' are valid. To that end, they adopt an alternative, efficient method for natural memorization. Using that, they test the two main findings of artificial memorization: that longer training and larger models increase memorization. In a set of experiments, they empirically find the opposite trend.

**Strengths:**

- The topic of memorization is highly relevant not only to address data privacy, but also to get a better understanding of generalization and effective model design.
- The submission is generally well-written, identifies a clear question and then takes a straight path to answering it.
- The experiments are well-suited to test the findings of previous work (size and training length correlate with memorization).
- The findings are interesting, and spark new questions for further research.

**Weaknesses:**

- As far as I can tell, the submission has no original method contribution, it re-uses definitions for and methods to measure memorization from previous work. It presents the empirical test if findings from artificial proxies hold in settings for natural memorization.
- Further, the limited experiments also limit the empirical significance of the submission. To me, it seems that the experiments show the necessary signal to dis-prove the abovementioned correlations of model size and training duration with memorization, but stop there. I would encourage the authors to test the generality of their findings, e.g. by varying architectures more systematically (in both depth and width), changing the dataset and task. Their test setup may also allow to vary a broader range of hyperparameters that do affect performance and generalization gap, s.t. they can test the relation to memorization on a broader basis. Lastly, I would appreciate a test of how far the findings carry over to realistic training regimes, e.g. with data augmentation.

I encourage the authors to continue work on the submission, and provide broader experiments and insights into what could drive memorization or how their results, e.g. on generalization gap, might be reconciled with existing work. As it currently is, I believe it does not suffice to recommend acceptance.

**Questions:**

No further questions, please refer to weaknesses.

---

> ### Author Response · Authors · 2024-11-19
>
> > generality of their findings
>
> We thank the reviwer for their comments. To increase the generality of our findings, we repeated our experiments on a more complex dataset, Tiny ImageNet, which has 200 classes over 100,000 images using both ViT based and Resnet based architectures. We observed that our earlier findings still hold. This demonstrates that our results are independent of model architecture and pertaining. We have updated the paper to reflect the changes.
>
> > data augmentation
>
> Our current training pipeline employs data augmentation techniques such as Random Horizontal Flip and Random Translate to train high-quality models. The reported results are based on this augmented training pipeline. In earlier versions of our experiments, we trained models without augmentation and observed a larger train-test gap, which led to increased memorization. However, we did not include these results since test accuracy without augmentation was low, which could have raised reviewer concerns.

---

> > ### Comment · Reviewer_JPiP · 2024-11-22
> >
> > I would like to thank the authors for their response and the continued work on their submission. I am not entirely sure what revision has changed, but I assume current Tables 1 and 2 and Figures 1, 2, 3 and 4 are new?
> >
> > While I appreciate the replications of the experiments on different and larger architectures and more datasets, I am wondering if there are specific effects that result in the particular outcome. As the authors note, data augmentation is necessary to achieve small train-test gap, I imagine particularly for the larger architectures. I assume this explains that larger architectures generally improve in test performance, rather than overfit more.
> >
> > This leads me to wonder whether the visualizations over the different architectures and model sizes are - in fact - effects of different train-test gaps? In that case, the author's findings for decreasing memorization while increasing model size may hold only for very well parameterized training regimes. Likewise for training duration.
> >
> > I would be curious to understand the different effects better. For example, the authors may try to disentangle model size from train-test gap by varying the training setup as well, s.t. model size and train-test gap vary independently.

---

> > > ### Author Response · Authors · 2024-11-22
> > >
> > > We would like to sincerely thank the reviewer for engaging with us in this discussion.
> > >
> > > ### Updates to the Paper
> > > Table 2, Fig. 1(b), Fig. 2(c), Fig. 3 (ImageNet section), and Fig. 4 are new additions.
> > >
> > > ---
> > >
> > > > **I assume this explains that larger architectures generally improve in test performance, rather than overfit more.**
> > >
> > > Overfitting does occur in larger architectures. However, since larger architectures are capable of learning more compared to smaller ones, we observe less memorization in larger models.
> > >
> > > ---
> > >
> > > > **May hold only for very well-parameterized training regimes.**
> > >
> > > We are not entirely sure what the reviewer means by “well-parameterized” training regimes. If the term refers to *over-parameterization* (i.e., having more parameters than training points), our findings contradict the current literature, which is often based on artificial memorization studies.
> > >
> > > If the question is about whether our findings hold for *under-parameterized* networks, we believe this might not be a meaningful avenue of exploration. This is because most training regimes prioritize over-parameterization (i.e., having more parameters than training points) to maximize test accuracy. For instance, the smallest model we could identify with high accuracy for CIFAR-10 has **0.17M weights**, which is approximately 3.5 times the size of the dataset.
> > >
> > > ---
> > >
> > > > **Model size and train-test gap vary independently.**
> > >
> > > Our results provide some insights into this question. For example, consider the Tiny ImageNet experiments. As shown in Table 2, ResNet-50 and ViT-Small have similar numbers of model parameters. However, ViT-Small memorization a smaller fraction of the data than ResNet-50. This highlights that while depth is *necessary*, it is not a *sufficient* condition for reducing memorization.

---

> > > > ### Comment · Reviewer_JPiP · 2024-11-25
> > > >
> > > > I thank the authors for their response. With well-parameterized, I referred to the training setup including regularization and augmentations.
> > > >
> > > > I appreciate the comparisons of different models with the same training setup, but I'm not sure how much can be inferred from the comparisons the authors make. My point is, that I am sure there is a training setup without regularization and augmentation, in which deeper models and longer training causes a larger train-test gap. The authors appear to indicate that in their previous response.
> > > > > In earlier versions of our experiments, we trained models without augmentation and observed a larger train-test gap, which led to increased memorization.
> > > >
> > > > If training / experiment setup can practically invert the findings, I think they should be considered or controlled for.
> > > >
> > > > Therfore, I encourage the authors to vary the training setup s.t. the same model varies in performance and train-test gap to disentangle the current candidates for causes of memorization. I believe this may help build solid understanding as to whether or not depth, training duration, train-test gap or something else are indicators for or root causes of memorization.

---

> ### Author Response · Authors · 2024-11-25
>
> Thats a fair question. To answer it, we followed reviewer suggestion and we repeated our experiments for CIFAR-10 without data-augmentation and weight decay. Our findings (in Figure 4 in the Appendix) remain consistent with the earlier experiments. a) Increasing model complexity decreases memorization b) Increasing iterations decreases memorization c) Memorization and Train-Test gap are strongly correlated (Pearson score 0.99).
>
> This shows that augmentation and weight decay are not responsible for findings.
>
> Please let us know if you have any more questions or if you have suggestions on how we can improve our work.

---

> > ### Author Response · Authors · 2024-11-25
> >
> > Dear Reviewer JPiP,
> >
> > Thank you for your valuable feedback. We have provided detailed responses to your questions and hope they address your concerns thoroughly. As the interactive discussion period is coming to a close, please don’t hesitate to reach out if you need further clarification or additional information.

---

> > > ### Author Response · Authors · 2024-11-30
> > >
> > > Dear Reviewer,
> > >
> > > As we near the discussion deadline, please let feel free to let us know if you have any more questions or suggestions.

---

### Official Review · Reviewer_fTGZ · 2024-11-01

**Soundness:** 3
**Presentation:** 3
**Contribution:** 3
**Rating:** 6
**Confidence:** 4

**Summary:**

The paper focuses on memorization in supervised classification models. The author state that it is often difficult to measure real/natural memorization since the only methods available rely on removing specific data points from the training set to be able to study wether this specific data point is memorized or not by the model. Intuitively, if the model is trained on enough data, removing any given data point should not impact the model's prediction over this data point if the model was able to generalize. However, if removing the data point induces a wrong answer, it is often considered that this data point could only have been memorized by the model. Such setup is often costly, so researchers often use proxies (coined as artificial memorization) to avoid training thousand of models. Those proxies mostly rely on the addition of noise to approximate natural memorization phenomena. Based on the literature on artificial memorization, it is often considered that increasing the model size or the number of epochs lead to a greater risk of memorization. The authors challenge this hypothesis by shedding light on the relationship between the double descent phenomena (in which the test error rate can start to get lower after having increased) with natural memorization. They especially show that by training longer and with more capacity, the memorization can actually decrease after having increased significantly (which seems to be in contraction with the finding from the artificial memorization literature).

**Strengths:**

- The paper is well motivated and well written.
- The authors introduce the interesting concept of "transient memorization" about data points that can be memorized under certain models by not by others. This is interesting to know that smaller models will memorize those point easily, while bigger models will generalize to them.

**Weaknesses:**

- The abstract start by stating that memorization is the ability of "deep learning models to assign ground truth labels to inputs". I would be cautious about such statement since a generative model that do not learn from any ground truth labels, could memorize its inputs simple because the model is just train to reconstruct the input.  So I would definitively not constrained the definition of memorization in deep learning to having ground truth labels. Instead, I would recommend the authors to specify directly that they are interested in supervised scenarios to avoid any misunderstanding.
- Not really sure that Figure 1 is even needed in the first place, most of the readers will know what unstructured noise is.
- Incorrect statement. The authors wrote "Double descent experiments have been conducted on artificial data ... " Some double descent experiments like in 1) have been conducted on dataset like cifar10.
- Experiments are being conducted on only two very similar datasets CIFAR10/CIFAR100. Even if the compute cost can justify such limited experiment setup, I would be more cautious about the claim being made (such as "we show that this understanding is incorrect", "Those does not apply..."). I would advise to downplay a bit them by specifying something like "Given our experimental results on CIFAR10 and CIFAR100, we challenge the idea that artificial memorization might always be a good proxy for...". And maybe add as future work, that more experiments might be needed to confirm those results.

1) Multi-scale Feature Learning Dynamics: Insights for Double Descent, Pezeshki et al. ICML 2022

**Questions:**

- Any insights on the relationship between weights-decay and memorization. Since increasing the model's capacity can reduce memorization, do having a stronger weight decay could increase memorization?
- Are your results still holding when using data augmentations?

---

> ### Author Response · Authors · 2024-11-19
>
> We thank the reviewer for their thoughtful comments. We have incorporated *all* the suggested changes from the review.
>
>
> **Experiments are being conducted on only two very similar datasets, CIFAR10/CIFAR100.**
>
> We appreciate this point. In response, we have extended our experiments to include the Tiny ImageNet dataset (200 classes, 100,000 data points) using both ResNet and ViT families of models. Our findings remain the same across these additional experiments. We have updated the manuscript to reflect these new experiments. Additionally, as recommended, we have softened the wording in the paper to better align with the scope of the findings.
>
>
> **Any insights on the relationship between weight decay and memorization?**
>
> Based on our experience, the best way to conceptualize the relationship between any single component and memorization is through the lens of the train-test gap. A larger gap typically indicates higher memorization. Excessive weight decay likely reduces both training and test accuracy, resulting in a model that neither learns effectively nor memorizes. However, as weight decay increases up to a certain point where it widens the train-test gap, it is more likely to promote memorization.
>
>
> **Do your results still hold when using data augmentations?**
>
> Yes! Our current training pipeline employs data augmentation techniques such as Random Horizontal Flip and Random Translate to train high-quality models. The reported results are based on this augmented training pipeline. In earlier versions of our experiments, we trained models without augmentation and observed a larger train-test gap, which led to increased memorization.

---

> > ### Author Response · Authors · 2024-11-25
> >
> > Dear Reviewer fTGZ,
> >
> > Thank you for your valuable feedback. We have provided detailed responses to your questions and hope they address your concerns thoroughly. As the interactive discussion period is coming to a close, please don’t hesitate to reach out if you need further clarification or additional information.

---

> > > ### Comment · Reviewer_fTGZ · 2024-11-26
> > >
> > > Thank you for answering my concerns and updating the paper. The tiny imagenet experiments are a nice addition to the paper.
> > > I will just advise to put only the results of CIFAR10 and Tiny-Imagenet in Figure 2 (and leave CIFAR-100 for the appendix).

---

> > > > ### Author Response · Authors · 2024-11-28
> > > >
> > > > Done. Thank you for your feedback. Please do not hesitate to let us know if there are any other ways with which we can improve our work.

---

> > > > > ### Author Response · Authors · 2024-11-30
> > > > >
> > > > > Dear Reviewer,
> > > > >
> > > > > As we near the discussion deadline, please let feel free to let us know if you have any more questions or suggestions.

---

### Author Response · Authors · 2024-11-20
**Summary of Changes**

We sincerely thank all the reviewers for their feedback. We have incorporated their suggestions in the form of  additional experiments and textual changes listed below:

Abstract modified. **[fTGZ]**

Figure 1 removed. **[fTGZ]**

Incorrect statement fixed.**[fTGZ]**

Downplayed sentences in conclusion and added future work direction. **[fTGZ]**

Added Sentence on data augmentation used during training. **[fTGZ, JPiP]**

Additional Experiments on Tiny Imagenet and ViT models **[fTGZ, JPiP,C2wk]**

Reworked Transient Memorization into two parts (Temporal-wise and Model-wise) **[C2wk]**

Show distribution of memorization scores (Figure 4) **[C2wk]**

Show Train-test gap vs memorization over Tiny Imagenet and transformer models (Figure 3) **[C2wk]**

Future work to evaluate findings over LLMs **[SHMd]**

Our findings contradict Feldman et al. **[SHMd]**

---

### Meta-Review · Area_Chair_jrZn · 2024-12-23

**Metareview:**

The paper provides some evidence that prior assumptions about the relationship between memorization, model size, and training duration may not hold when tested on real data, since most prior works dealt with artificial memorization. While the submission makes an effort to identify relevant effects, drawing definitive conclusions beyond the known correlation between the generalization gap and memorization remains challenging. There are numerous potential confounding factors that could influence the results, which would need to be systematically tested—ideally disentangled from the generalization gap—to establish clear relationships, such as between model size and memorization. During the rebuttal discussions, authors highlight the importance of the experimental setup in supporting their hypothesis. While the approach shows promise, it may require a deeper understanding of the robustness and underlying causes of the observed effects. Thus, the current decision is Reject, but AC encourages the authors to incorporate the reviews for an updated manuscript and submit it in a future venue.

**Additional Comments On Reviewer Discussion:**

Reviewers actively engaged in the rebuttal process. All reviewers commended extensive experimental work of the paper.

SHMd mainly pointed out that the methodological novelty is weak and the paper fails provide more analytic intuition of the method.

JPiP mentions "the submission makes efforts in identifying effects, but other than the known correlation between generalisation gap and memorization, I find it challenging to draw definitive conclusions.", and it is really challenging to find the real cause, since there are too many confounding factors.

C2wk originally gave a positive review, but later during the rebuttal, he/she agreed to SHMd and JPiP.

---

### Decision · Program_Chairs · 2025-01-22

Reject